# Automatic Tunnel Steel Arches Extraction Algorithm Based on 3D LiDAR Point Cloud

**DOI:** 10.3390/s19183972

**Published:** 2019-09-14

**Authors:** Wenting Zhang, Wenjie Qiu, Di Song, Bin Xie

**Affiliations:** School of Automation, Central South University, Changsha 410083, China; zkxdoit@csu.edu.cn (W.Z.); qiuwenjie1027@163.com (W.Q.); sd3_skr@163.com (D.S.)

**Keywords:** 3D LiDAR point cloud, boundary detection, region-growing, tunnel

## Abstract

Automation is an inevitable trend in the development of tunnel shotcrete machinery. Tunnel environmental perception based on 3D LiDAR point cloud has become a research hotspot. Current researches about the detection of tunnel point clouds focus on the completed tunnel with a smooth surface. However, few people have researched the automatic detection method for steel arches installed on a complex rock surface. This paper presents a novel algorithm to extract tunnel steel arches. Firstly, we propose a refined function for calibrating the tunnel axis by minimizing the density variance of the projected point cloud. Secondly, we segment the rock surface from the tunnel point cloud by using the region-growing method with the parameters obtained by analyzing the tunnel section sequence. Finally, a Directed Edge Growing (DEG) method is proposed to detect steel arches on the rock surface in the tunnel. Our experiment in the highway tunnels under construction in Changsha (China) shows that the proposed algorithm can effectively extract the points of the edge of steel arches from 3D LiDAR point cloud of the tunnel without manual assistance. The results demonstrated that the proposed algorithm achieved 92.1% of precision, 89.1% of recall, and 90.6% of the F-score.

## 1. Introduction

In recent years, the demand for intelligent construction machinery used in a harsh environment has been constantly increasing [1]. In the process of shotcrete on the inner surface, since there is low visibility due to the presence of dust (Figure 1a), intelligent shotcrete machinery is urgently needed to totally replace manual work. Therefore, it is necessary to conduct three-dimensional (3D) reconstruction and feature detection algorithms for the shotcrete environment of the tunnel. Initial shotcrete has been used on the rock surface [2], which is the surface with rock masses of the tunnel after blasting. As shown in Figure 1b, multiple supporting structures, such as steel arches and steel wire mesh, may be installed on the rock surface [3], whereas the other two kinds of surfaces (shotcrete surface and working surface) are smoother with simple structures. Therefore, the rock surface is one of the most complex sections in the tunnel surface point cloud (Figure 1c) to realize 3D reconstruction and detection. To ensure that the sprinkler can be perpendicular to the surface in the process of shotcrete, the orientation of the sprinkler should be adjusted at the edge of the steel arches (Figure 1d). Therefore, the position of the steel arches needs to be extracted in advance. Since the steel arch area has continuous geometric characteristics, it can be considered as a 3D edge feature according to the definition of the 3D edge of Ni et al. [4]. In Figure 2, a longitudinal section is shown to be taken from the tunnel point cloud along the tunnel direction to show the local structure of the steel arch more clearly.

The structure of the longitudinal section is shown in Figure 3. The point cloud of the steel arch areas (the red points in Figure 3) has unique characteristics, which are different from other kinds of feature lines, such as the road edge [5,6], building outline [7,8], and components edge [9]. The most significant geometric feature of steel arches is that it is a standard I-steel with a smooth surface and a regular arch corner. Besides, the outline of the steel arch varies with the thickness of the concrete cover, and the outline of the exposed steel arches are clearer than that of the steel arches covered by concrete. Since the point cloud of each arch area is a continuous line of the point cloud (as shown in Figure 1d and Figure 3), the bottom edge of the arch can be viewed as an exterior boundary [10] of the point cloud.

### 1.1. Related Works

***Research related to tunnel axis:*** In current research, the LiDAR is usually set at an uncertain position in the tunnel in the scanning process. Therefore, obtaining the tunnel line, which is parallel to the tunnel axis in every position of the tunnel, is a common problem. Researchers generally obtain the tunnel line first and then analyze the tunnel profiles perpendicular to the line. The papers [11,12,13] specialized in extracting tunnel axis and splicing the tunnel point cloud. Zhu et al. [14] extracted the tunnel line by fitting one circular surface. Kang et al. [13] extracted the tunnel line by obtaining the middle line parallel to both sides of the tunnel. However, their methods are only suitable for round tunnels with a smooth surface, which are highly sensitive to the shape of the tunnel. They are not suitable for tunnels with multiple types of shapes and those with a large number of point clouds missing. Zhou et al. [15] extracted the track line based on 3D linear fitting for the rail tunnel point cloud after slicing, to obtain the central line of the tunnel based on the track line. The segmented projection method proposed by Gonçalves et al. [16] was sensitive to discrete points, and manual adjustment of outliers was required. The tunnel sections can be used to detect the profile of the tunnel section, but it is not enough to complete more flexible 3D feature recognition. Han et al. [17] believed that there was uncertainty in obtaining 2D tunnel profile through step-size projection of 3D data, which could result in the information loss of 3D LiDAR data. Therefore, we proposed an axis extraction algorithm based on Rotational Projection Density Variance (RPDV) to detect the tunnel axis not based on fitting the overall geometrical outline of the tunnel. Contrary to the above method, the proposed algorithm has good robustness to the tunnel profile shape.

***Research related to rock surface***: In order to reduce computational complexity and improve the steel arches extraction accuracy, the rock surface should be extracted in advance. Related research focuses on the volume calculation of the overbreak and the blasting characteristics of the rock surface [2,18,19]. The research about tunnel feature detection focus on the underground surface monitoring and the deformed cross-section of the completed tunnel [11,15,16]. For the treatment of disordered, rough, and multiple types of point clouds, researchers usually visualize and store them for post-processing by interactive software [16,20,21,22], which is not suitable for automated construction scenarios. Lai et al. [21] proposed a new method for building an index of a point cloud of multiple tubular types of curved surfaces similar to a tunnel. They remolded the 3D grid of a mine roadway and projected it onto a 2D plane to output a uniform grid index of a tunnel point cloud [21]. This method requires user interaction to achieve a more uniform rasterization effect. Previous research on rock surfaces rely on man-machine interaction and are mainly used in post-processing. We proposed Differential Analysis for the Section Sequences of the Tunnel point cloud (DASST) method for rock surface segmentation to narrow the search area of steel arches, which applies to the construction site without manual parameter regulation.

***Research related to tunnel supporting structure:*** Former research has usually studied the rock surface of the tunnel or the lining surface, while other structures installed manually in tunnels are generally removed as noise. Cheng et al. [23] designed an angle-based morphology-related filtering method, which automatically and indiscriminately removing all non-lining points from non-circular tunnel sections. Mah et al. [24] explored the removal of steel mesh in tunnel surface data, and adopted the Principal Component Analysis (PCA) method, analyzing the local area of the network without considering the shape type of the whole tunnel. These previous studies on tunnel cross-section detection usually adopted a semi-automatic method, which requires manual adjustment and modification of algorithm parameters in the processing process. Elberink et al. [25] researched the automatic extraction algorithm of multiple parallel rail tracks, which had similar outlines with the tunnel steel arches. The rail tracks are built on the flat ground, while the tunnel steel arches are installed around the tunnel walls. Du et al. [26] proposed a gradient statistical method to determine the mileages of the circular segment joints by the histogram statistical method and the peak detection method. This method relies on selecting appropriate angular intervals with fewer appendages to improve the detection accuracy [26]. Since there are rare researches about steel arches extraction, our research has innovative value in regard to the theories and current practices. The Directed Edge Growing (DEG) method using the local normal saliency as the evaluation index was particularly proposed for extracting steel arches from the rock surface.

### 1.2. Contributions

We studied the problem of extracting steel arches from a 3D LiDAR point cloud of variable cross-section tunnels, such as round tunnels used for transportation and square tunnels used in coal mines. We designed a novel steel arch extraction algorithm, including three steps based on the features of the tunnel.

In the following, we calibrate the tunnel axis of the point cloud in the world coordinate system by minimizing the Rotational Projection Density Variance (RPDV), which is profile-insusceptible to the tunnel; then, we propose an adaptive threshold extraction algorithm to extract the rock surface by using the Differential Analysis for the Section Sequences of the Tunnel point cloud (DASST); and finally, we propose a Directed Edge Growing (DEG) method to extract the steel arches edge from the rock surface, using the local normal saliency as the evaluation index of candidate points.

## 2. Data Collection and Dataset

### 2.1. Data Acquisition Scenario

The proposed algorithm was tested on a series of real datasets, including nine sets of tunnel point clouds collected by the researchers in a road tunnel under construction in Changsha, Hunan province, China. The tunnels scenario had just been blasted and installed with the supporting structures, including steel arches. There were some limitations of data acquisition, as follows:The tunnel was of low visibility due to the dusty air and lack of lighting conditions.The 360 degree view or multiple view registration was needed for data scanning.During data collection, the steel arch would inevitably block the rear tunnel wall and result in missing data.In the tunnel construction site, non-staff were rarely allowed to enter, and the residence time was limited to avoid delaying the construction progress.

### 2.2. Acquisition Equipment

Traditional tunnel detection methods use the total station to locate several measurement points on the steel arches [27]. Currently, some researchers use LiDAR (Light Detection and Ranging) and RGB cameras for tunnel inspection [28]. Considering the working environment introduced in Section 2.1, we selected LiDAR as the acquisition equipment by comparing the three instruments in Table 1.

In this paper, a 2D LiDAR electronically controlled by a motor with a low cost was selected to scan and reconstruct the tunnel environment. Table 2 describes the 3D scanning device. The absolute accuracy of the scanning device for the distance of 30 m was within 25 mm.

The scanning device was set on a metal holder, which was randomly placed on the uneven ground in the tunnel (Figure 4a) to collect the data set. In practice, the LiDAR is designed to be precisely installed on the shotcrete machine (Figure 4b). The scanning data was used to realize the measurement and perception of the tunnel environment for the shotcrete machine. Shotcrete machines are generally oriented toward the working surface of the tunnel, and there is no need for precise parking locations. In the scanning process, the mechanical arm stops moving, and one group of the original point cloud, shown as Figure 5, will be obtained after the rotary table has been rotated once. The coordinate system in Figure 5 is the right-hand coordinate system. The location of the point cloud relative to the shotcrete machine is obtained by the installation position and the kinematics solution of the mechanical arm.

### 2.3. Pre-Processing of the Data

Voxel filtering and density-based spatial clustering of applications with noise (DBSCAN) [29] were used in data preprocessing to de-noise the original data.

3D scanning of tunnels usually results in large amounts of point cloud data. Each group of this data has around millions of unevenly distributed points. In order to ensure that the point cloud had high density and uniform distribution, voxel filtering was first used with appropriate voxel size, determined to be 26 mm according to the resolution (0.05${}^{\circ }$) and scan range (30 m) of the sensor.
(1)$$\mathrm{voxelsize}=30000\tan0.05^{\circ }\approx 26\;\left(\mathrm{mm}\right)$$

There were people, equipment, and other shielding in the tunnel, resulting in the miscellaneous points and holes of the original point cloud (Section 2
Figure 5). Therefore, the DBSCAN method was adopted to remove the miscellaneous points and retain $P_{tun}\left(R_{n\times 3}\right)$, the point cloud of the main part of the tunnel, which includes the shotcrete surface, the rock surface, and the working surface. In the DBSCAN method, firstly the threshold of the searching radius should be set for Kd-tree searching. To avoid the point cloud of the steel arches from being removed, the searching radius should be set larger than the thickness of the arches.

For a random seed point *p*, the points within the searching radius could be found to form a new class $Q_{1}$. Then, each point in class $Q_{1}$ was used as a seed point to search more points within the radius threshold to expand $Q_{1}$, until $Q_{1}$ can no longer add new points. For the remaining points in the original point cloud, more classes $Q_{i}\left(i=2,3,\ldots ,n\right)$ were obtained by repeating the same steps. The number of clusters *n* is not necessary to specify in advance. Each new cluster starts at a randomly assigned seed point in the remaining set of points and stops when there are no remaining points relevant to the radius threshold. Then, $P_{tun}\left(R_{n\times 3}\right)$ was the largest class in $Q_{i}$.

### 2.4. Data Set and Basic Parameters

Figure 6 illustrates the parameters of the tunnel point cloud dataset. The parameter $W_{a}$ is the width between neighbouring arches, and *B* is the thickness of the arches.

Table 3 describes the tunnel point cloud data set with nine groups of the original point cloud collected in Changsha (China).

When data collection was carried out on the new working surface, the last working surface was completely covered by concrete, and the appearance was greatly changed. A different group of data was collected in a different time and space, and each group of data was meaningful only to the current temporary work surface. Therefore, different from the tunnel visualization and detection methods with multiple measurement points that require registration [30], registration was not required between groups of point clouds collected in this dataset. The observation location of the data was used for the data sorting number.

## 3. Methodology

### 3.1. Overview of the Proposed Algorithm

The flowchart of the proposed steel arches extraction algorithm is shown in Figure 7. For the pre-processed tunnel point cloud in Table 3, firstly, an evaluation index RPDV (Rotational Projection Density Variance) was proposed to represent the dispersion degree of the tunnel point cloud projection. By changing the projection angle to minimize the dispersion degree, the optimal angle of tunnel axis correction was determined. In order to extract the rock surface from the tunnel point cloud, we sliced the tunnel point cloud along the X-axis and used DASST to obtain an adaptive curvature and height threshold. A curvature threshold was used with the region-growing to extract the rock surface of the tunnel, and the height threshold was used after DBSCAN to remove the miscellaneous points on the left and right walls of the tunnel. Then, the DEG method was adopted based on the local normal saliency to extract and to fill steel arch edges.

### 3.2. Orientation Calibration

The tunnel axis is the most commonly used reference line in tunnel construction and measurement, and it is necessary to extract the tunnel axis first. Since the location of the LiDAR relative to the tunnel is unknown, we first adjusted the origin position and tunnel axis of the original point cloud, and normalized all point cloud data, as shown in Figure 8a. The tunnel bending could be ignored as the length of the construction section is short. Since the tunnel had similar characteristics to the tensile surface, the tunnel axis problem could be regarded as the problem of obtaining the stretching direction of the drawing surface.

Ke et al. [31] promoted an orientation correction algorithm of minimizing the projective area of the point cloud projected onto a plane, which was perpendicular to the stretching direction. The projective area was calculated by summarizing the number of the effective cells $N_{vox}$ (the squares occupied by the projected point cloud in Figure 8b). As shown in Figure 8b, when the *X*-axis has a particular inclination, the projected point will form a profile with a specific bandwidth. As shown in Figure 8c, if the *X*-axis is parallel to the tunnel axis of the tunnel, the projected point cloud on the Y-Z plane will theoretically form the outline of the tunnel.

Since $N_{vox}$ was interfered by the holes and the working surface of the tunnel point clouds, the method of Ke et al. [31] was not suitable for calibrating the tunnel point cloud collected in Section 2. When the *X*-axis approach was taken to the tunnel excavation direction, the projected point cloud was more concentrated near the theoretical profile. Therefore, the Rotational Projection Density Variance (RPDV) was proposed for calibrating the *X*-axis by minimizing the density variance of the projected point cloud.

Firstly, using the pass-through filter in the *Z*-axis direction and starting from the lowest point, the point cloud with a certain height was taken out, and the least square plane was fitted to get the plane equation of the ground. The height of the ground point cloud could take 500 mm, empirically. Then, the *Z*-axis of the coordinate system was calibrated to be perpendicular to the ground. RPDV method proposed in this paper was used to rotate the tunnel point cloud around the *Z*-axis until the *X*-axis (tunnel axis) was parallel to the tunnel orientation.

Projection matrix: Firstly, we calculated the center of mass $p_{c}\left(\bar{x},\bar{y},\bar{z}\right)$ of the tunnel point cloud $P_{tun}$ and shifted the coordinate origin of $P_{tun}$ to $p_{c}$. $M_{x}$ in Equation (Equation 2) was the projection matrix to project $P_{tun}$ onto the Y–Z plane.
(2)$$M_{x}=\left[\begin{array}{ccc} 0 & 0 & 0\\ 0 & 1 & 0\\ 0 & 0 & 1 \end{array}\right]$$Rotation matrix: Let $M_{r}$ be the rotation matrix that relates a certain vector $v_{0}$ in $R^{3}$, and the corresponding rotated vector is ${v_{0}}^{{}^{\prime }}=M_{r}v_{0}$. Given the rotation angle θ and the rotation axis n(u,v,w) (∥n∥=1), according to the rotation formula in the paper [32], the vector ${v_{0}}^{{}^{\prime }}$ can be represented as Equation (Equation 3), and the geometrical interpretation is shown in Figure 9. The rotation matrix $M_{r}\left(n,\theta \right)$ can be obtained from Equation (Equation 4).
(3)$${v_{0}}^{{}^{\prime }}=\left(1-\cos\theta \right)\left(n\cdot v_{0}\right)n+v_{0}\cos\theta +\left(n\times v_{0}\right)\sin\theta$$
(4)$$M_{r}\left(n,\theta \right)=\left[\begin{array}{ccc} u^{2}\left(1-\cos\theta \right)+\cos\theta & uv(1-\cos\theta )-w\sin\theta & uw(1-\cos\theta )+v\sin\theta \\ uv(1-\cos\theta )+w\sin\theta & v^{2}\left(1-\cos\theta \right)+\cos\theta & vw(1-\cos\theta )-u\sin\theta \\ uw(1-\cos\theta )-v\sin\theta & vw(1-\cos\theta )+u\sin\theta & w^{2}\left(1-\cos\theta \right)+\cos\theta \end{array}\right]$$Voxelized point cloud: Based on the thickness of the tunnel wall installed with the arches, we set *B* as the leaf size of voxelization. The unit vector parallel to the Z-axis is $v_{z}\left(0,0,1\right)$. We controlled the rotation step of the variation angle θ, which determines the calibration accuracy and time for computation. Then, the projected and voxelized point cloud of $P_{tun}$ is $V_{tun}\left(\theta \right)$.
(5)$$V_{tun}\left(\theta \right)=voxel\left(P_{tun}M_{r}\left(v_{z},\theta \right)M_{x}\right)$$The density variance of the point cloud: For a random voxelized point cloud V, we defined voxel(V) as the function to voxelize V. During the sampling process, the number $N_{vox}$ of the effective cells and the number $n_{i}\left(i\in \left[1,N_{vox}\right]\right)$ of points contained in each effective cell were obtained. Then, the Projection Density Variance (PDV) of V was defined as f(V) as in Equation (Equation 6).
(6)$$f\left(V\right)=\frac{\sum_{i=1}^{N_{vox}}{n_{i}}^{2}}{N_{vox}}$$The optimum angle $\theta_{m}$: With the rotation of θ, we obtained the optimum angle $\theta_{m}$ in the Rotational Projection Density Variance (RPDV) (Equation (Equation 7)) and the calibrated point cloud $P_{tuns}$ in Equation (Equation 8).
(7)$$\theta_{m}=\arg\max f\left(V_{tun}\left(\theta \right)\right)$$
(8)$$P_{tuns}=P_{tun}M_{r}\left(v_{z},\theta_{m}\right)$$As shown in Figure 10, the projection density follows the change of θ and shows a periodicity of 360. Due to the specific shape characteristic of the tunnel point cloud as a tensile surface, there is usually only one obvious optimal solution when θ changes within a range of 360, and the optimal angle values are rarely affected by a small number of outliers.

Since the X-axis of the LiDAR was initially directed to the working surface as illustrated in Section 2, Figure 5, the deviation angle θ was set to $\left(\theta \in \left[-\frac{\pi }{2},\frac{\pi }{2}\right]\right)$. According to the method, the ideal effect could be obtained by the tunnel axis calibration of the tunnel point cloud, as shown in Figure 11.

In addition, tunnel orientation calibration was used to realize tunnel slicing analysis and provide basic orientation for steel arch extraction. It is an intermediate link and will not bring rigid movement and error to the final result. After the post-processing, the transformed point cloud would be reversely converted back to the original coordinate system.

### 3.3. Extraction of Rock Surface

#### 3.3.1. Curvature of the Point Cloud

The rock surface is the region of interest where the steels are installed. Convexity is the most distinctive feature among three kinds of surfaces (Figure 1b). In current 3D point cloud processing research, Principal Component Analysis (PCA) [33] is commonly used to estimate the normal and curvature in each point of the point cloud. The research of Cao et al. [34] showed that PCA took the shortest time compared with other normal estimation methods. Therefore, PCA is suitable for large-scale point clouds. In this paper, we used the PCA method to process $P_{tuns}$. First, we searched all points in the neighborhood of a point *p* within radius $W_{a}$ and got *k* points. Then, the center point $p_{c}\left(\bar{x},\bar{y},\bar{z}\right)$ of $P_{tuns}$ was calculated. The covariance matrix $C_{o}$ was obtained from Equation (Equation 9).
(9)$$C_{o}=\frac{1}{k}\sum_{i=1}^{k}\left[\begin{array}{ccc} {\left(x_{i}-\bar{x}\right)}^{2} & \left(x_{i}-\bar{x}\right)\left(y_{i}-\bar{y}\right) & \left(x_{i}-\bar{x}\right)\left(z_{i}-\bar{z}\right)\\ \left(x_{i}-\bar{x}\right)\left(y_{i}-\bar{y}\right) & {\left(y_{i}-\bar{y}\right)}^{2} & \left(y_{i}-\bar{y}\right)\left(z_{i}-\bar{z}\right)\\ \left(x_{i}-\bar{x}\right)\left(z_{i}-\bar{z}\right) & \left(y_{i}-\bar{y}\right)\left(z_{i}-\bar{z}\right) & {\left(z_{i}-\bar{z}\right)}^{2} \end{array}\right]$$

Multiple eigenvectors and eigenvalues of the covariance matrix were calculated. Curvature was computed by the minimum eigenvalue in Equation (Equation 10). In addition, the eigenvector corresponding to the eigenvalue $\lambda_{min}$ is the normal vector N(p).
(10)$$curv=\left|\frac{\lambda_{min}\left({C_{o}}^{⊤}C_{o}\right)}{tr({C_{o}}^{⊤}C_{o})}\right|$$

#### 3.3.2. DASST Used for Region-Growing

The region-growing method is commonly used in point cloud segmentation. It obtains the boundary line of the surface at the end of the dividing surface and applies to the flat surface, such as buildings [35]. Therefore, the region-growing method was used to segment the tunnel surfaces according to the curvature. Firstly, if the curvature between the current seed point and the neighborhood point is less than the curvature threshold, it is classified into the same cluster. Then, two clusters with a curvature difference less than the curvature threshold were merged into the same cluster. The curvature threshold, which was the metric considered to measure the similarity in the region-growing method, could be obtained by analyzing the curvature characteristics of the tunnel.

Since the order of the three kinds of surfaces along the tunnel axis was determined, we proposed a Differential Analysis method for the Section Sequences of the Tunnel point cloud (DASST) to analyse the characteristics of the tunnel. We took sections with the slice thickness $W_{d}$ ($W_{d}\in \left[voxelsize,W_{a}\right]$) of $P_{tuns}$ along the X-axis, and the Section Sequence of the Tunnel point cloud (SST) $P_{i}\left(i=1,2,\ldots ,n\right)$ were obtained.

The parameter $\bar{curv}\left(i\right)$ was used to mean the curvature of all *k* points in the section $P_{i}$.
(11)$$\bar{curv}\left(i\right)=\frac{1}{k}\sum_{j=1}^{k}curv\left(P_{i}\left(j\right)\right),\left(i=1,2,\ldots ,n\right)$$

The discrete sliding cumulative function $g_{1}\left(i,t_{1}\right)$ with step size $t_{1}$ was proposed to obtain the curvature parameter $C_{m}$, which was used for region-growing.
(12)$$g_{1}\left(i,t_{1}\right)=\sum_{k=0}^{t_{1}}\left[\bar{curv}\left(i+k\right)-\bar{curv}\left(i\right)\right],\left(i<n-t_{1}\right)$$

In Equation (Equation 12), step size $t_{1}$ was set according to the structure of the tunnel. Under real working conditions, there are usually 2to3 arches located on the rock surface. To avoid the impact of the dramatical changing of the curvature between neighbouring arches, $t_{1}$ was set to $3W_{a}/W_{d}$, which is close to the width along side the X-axis of the rock surface installed with the arches. Assuming the optimal solution is $m_{1}$, then $C_{m}$ could be obtained in Equation (Equation 14).
(13)$$m_{1}=\arg\max_{1<m_{1}<n-t_{1}}g_{1}\left(m_{1},t_{1}\right)$$
(14)$$C_{m}=\frac{1}{m_{1}}\sum_{i=1}^{m_{1}}\bar{curv}\left(i\right)$$

The processing result of a group of real data is shown in Figure 12. As shown in Figure 12a, the average curvature of shotcrete surfaces were obtained by preliminarily separating shotcrete surfaces and rock surfaces by DASST. Since the cluster with the most points belonged to the shotcrete surface, the curvature was used as the threshold to remove the shotcrete surface from the tunnel point cloud by the region-growing method (Figure 12b). Since there were some miscellaneous points in the remaining point cloud (Figure 12c), DBSCAN was used to remove the largest surface and preserve the remaining point clouds $P_{tuns^{\prime }}$. As shown in Figure 12d, some redundant stone was piled up along the sides of the tunnel, connected with the rock surface, and could not be removed by the DBSCAN method. Therefore, the DASST method was used again to intercept the rock surface after DBSCAN processing.

From the DBSCAN-processed point clouds $P_{tuns^{\prime }}$, we obtained $n_{2}$ sections $W_{d}$ and calculated the mean height of each section $P_{i}$,

(15)$$h\left(i\right)=\frac{1}{k}\sum_{i=1}^{k}P_{i}.z\left(i\right),\left(i=1,2,\ldots ,n_{2}\right).$$

Similar to Equation (Equation 12), another discrete sliding cumulative function $g_{2}\left(i,t_{2}\right)$ with step size $t_{2}$ was proposed to obtain the segmentation position $m_{2}$ of the rock surface.

(16)$$g_{2}\left(i,t_{2}\right)=\sum_{k=0}^{t_{2}}\left[h(i+k)-h(i)\right],\left(i<n_{2}-t_{2}\right)$$

(17)$$m_{2}=\arg\max_{1<m_{2}<n_{2}-t_{2}}g_{2}\left(m_{2},t_{2}\right).$$

Actually, the height mutation between the rock surface and shotcrete surface of the tunnel usually occurs within the distance of $W_{a}$. Therefore, the step size $t_{2}$ was set to $W_{a}/W_{d}$. In Equation (Equation 18), all points at the right of slice $m_{2}$ on the direction of the X-axis form the point cloud $P_{rock}$, which is shown in Figure 13.

(18)$$P_{rock}=P_{tuns^{\prime }}\left\{P_{tuns^{\prime }}.x\geq \left(\max\left(P_{tuns^{\prime }}.x\right)-m_{2}t_{2}\right)\right\}$$

By selecting the maximum of functions $g_{1}$ and $g_{2}$, the corresponding independent variables $m_{1}$ and $m_{2}$ could be obtained. The purpose of obtaining the dividing point $m_{1}$ and $m_{2}$ was to remove the useless edge and segment the region of interest (the rock surface). Since the boundary between the shotcrete surface and the rock surface is obvious, the boundary position could be obtained by the proposed algorithm stably. Therefore, this paper does not further explore the accuracy of the numerical values of $m_{1}$ and $m_{2}$.

### 3.4. Extraction of Steel Arches

#### 3.4.1. Feasible Methods

For 3D point cloud feature recognition, traditional methods generally extract the feature points of point cloud and then use the geometric feature recognition method to extract the target areas from the feature points. There is a common and basic method of tunnel section detection based on profile tolerance [36] and tunnel axis [37], and several other methods can be considered for tunnel steel arch extraction, such as Harris3D [38], SIFT3D [39], and NARF [40] feature the point detection method. In addition, boundary detection [10] can be used to detect the points at the edge of the arch. Currently, there are some convolutional neural networks that can be used for point cloud classification and segmentation, mainly including: (a) multi-view based methods, such as MVCNN [41] and GVCNN [42]; (b) voxel-based methods, such as VoxNet [43], PointGrid [44]; and (c) point cloud-based methods, such as PointNet [45]. These methods do not apply to the arch extraction for the following common reasons:

(a) There is a relatively smooth transition surface between the steel arch and the surrounding tunnel wall, and the segmentation boundary is fuzzy. Theoretically, the current convolutional neural network is difficult to segment the arch edge accurately.

(b) The goal of steel arch detection is to obtain the outermost edge profile of yhe steel arch, which has prominent geometric characteristics and is suitable for traditional point cloud detection methods.

These feasible methods introduced above are compared in Table 4.

#### 3.4.2. Steel Arch Extraction Based on DEG

The region-growing method usually takes the point with the smallest curvature as the seed point, and then judges whether the candidate point in the neighbourhood belongs to the same region with the seed point. The restriction conditions of the region-growing method can be adjusted according to specific problems. Pan et al. [46] chose the boundary lines of overlapping regions instead of individual points as seeds of the region-growing method. Wang et al. [47] detected boundaries based on local normal saliency and extracted the lowest points.

The proposed algorithm of arch point extraction was based on the geometric characteristics of arches. Therefore, some restrictions were added to the location of seed points, the selection conditions of candidate points, and the growth direction of seed points. Compared with the region-growing, DEG has different retrieval constraints, as follows:The initial seed point is multiple, and distributed along the side of the point cloud. The Directed Edge Growing (DEG) method uses a line of seeds to extract points on the continuous bulges based on the region-growing method.The point at the lowest position in the neighborhood (edge point) is selected as the new seed point and stored in the arch feature set. The growth conditions of DEG are determined by the local normal saliency of the seed points. The most salient points in the local normal direction are searched in the candidate point neighbourhood within the searching radius, and stored in $P_{arch}$.Seed points should grow along a changing orientation of the local surface. When the neighbourhood point set is empty, seed points should be interpolated automatically.

This paper set the seed points along the edge of the tunnel point cloud and extracted the steel arch edge based on DEG. These seed points were used to conduct directed growing in point cloud $P_{rock}$, and the missing points of the arch were interpolated to obtain an accurate arch point cloud $P_{arch}$. Figure 14 shows the growing process and start–stop conditions of the seed points. The normal vectors N were calculated in Section 3.3.1.

As shown in Figure 14a, $L_{s}$$\left(L_{s}\left[k\right],k=1,2,\ldots ,n\right)$ is the point set of the initial points, which is set uniformly in a line along the edge of one side of the tunnel and parallel to the X-axis. The nearest point of each point in $L_{s}$ on the edge of the point cloud $P_{rock}\left[i\right]$ formed the initial seed point set $L_{1}$. The candidate $c_{i,j}$ was selected from the seed point set $S_{i}$ to search the optimum point $O_{i}$.

As shown in Figure 14b, the optimum point $O_{i}$ was used to reach the next seed point set along the direction vector set $v_{i}$. The normal vector set of $S_{i}$ is $N(S_{i})$, which was used to search $O_{i}$. The normal vector set of $O_{i}$ is $N(O_{i})$ and was used to calculate $v_{i}$. During the edge growing process, the starting and ending angle was $s_{1}$ and $s_{2}$, respectively.

For a point *i* in point cloud $P_{rock}$, $P_{rock}\left[i\right]\left(x,y,z\right)$ and $N(P_{rock}\left[i\right])$ was the three-dimensional coordinate information and the normal vector, respectively. When calculating the normal direction, the calculation results need to be adjusted to point inside the tunnel according to the dot product of $N(P_{rock}\left[i\right])$ and $P_{rock}\left[i\right]$, as shown in Equation (Equation 19).
(19)$$N\left(P_{rock}\left[i\right]\right)=sgn\left(N\left(P_{rock}\left[i\right]\right)\cdot P_{rock}\left[i\right]\right)N\left(P_{rock}\left[i\right]\right)$$

For point *p* in point cloud P, we searched all points within the radius of $W_{a}$ and obtained the point set $P_{p}i,i\in \left[1,n\right]$. Then, the local normal saliency of point $P{}_{p}i$ was obtained from Equation (Equation 20).
(20)$$D\left(P{}_{p}\left[i\right]\right)=\left(P{}_{p}\left[i\right]-p\right)\cdot N\left(p\right)$$

The process of obtaining each kind of key point is as follows:Initial point $L_{s}$As shown in Figure 14b, $s_{1}$ and $s_{2}$ were obtained by the Y and Z coordinate values of $P_{rock}$.
(21)$$s\left(P_{rock}\right)=\min_{P_{rock}}\frac{P_{rock}\left[i\right].z}{\left|[P_{rock}\left[i\right].y\;P_{rock}\left[i\right].z]\right|},P_{rock}\left[i\right]\in P_{rock}$$
(22)$$s_{1,2}=\arcsin s,\left(s_{1,2}\in \left[-\frac{\pi }{2},\frac{3\pi }{2}\right],s_{1}<s_{2}\right)$$A row of initial points $L_{s}$ were set uniformly at the starting position with the interval distance *B*.
(23)$$\left\{\begin{array}{cc} & L_{s}\left[k\right].x=\min_{P_{rock}}\left(P_{rock}.x\right)+B\cdot \left(k-1\right)\\ & L_{s}\left[k\right].y=\tan s_{1}\min_{P_{rock}}\left(P_{rock}.z\right)\\ & L_{s}\left[k\right].z=\min_{P_{rock}}\left(P_{rock}.z\right)\\ & k\in \left[1,\min_{P_{rock}}\left(P_{rock}.x\right)/B\right],k\in N^{*} \end{array}\right.$$Initial seed $L_{1}$$L_{1}$ were found by searching the nearest point in $P_{rock}$ by the Kd-tree method for every point in $L_{s}$. In addition, the initial seeds belong to the optimum points. That means
(24)$$O_{1}\left[k\right]=L_{1}\left[k\right].$$New seed $S_{i}$The direction vector $v_{i}\left[k\right]$ for $O_{i}\left[k\right]$ to reach a new seed point $S_{i}\left[k\right]$ was obtained by moving $O_{i}\left[k\right]$ along with the tunnel wall with the step size of 2B. The vector $v_{i}\left[k\right]$ and $S_{i}\left[k\right]$ were obtained from Equations (Equation 25) and (Equation 26), respectively.
(25)$$v_{i}\left[k\right]=2B\cdot \frac{v_{x}\times N\left(O_{i}\left[k\right]\right)}{|v_{x}\times N\left(O_{i}\left[k\right]\right)|}$$
(26)$$S_{i}\left[k\right]=O_{i}\left[k\right]+v_{i}\left[k\right]$$Since $S_{i}\left[k\right]$ is likely to be a point that does not exist in $P_{rock}$, its normal vector $N(S_{i}\left[k\right])$ was assigned by the normal vector of the nearest point $S_{i}{\left[k\right]}^{\prime }$ in $P_{rock}$.
(27)$$N\left(S_{i}\left[k\right]\right)=N\left(S_{i}{\left[k\right]}^{\prime }\right)$$Optimum point $O_{i+1}$All the candidate points $c_{i,j}$ were found by searching the points near $S_{i}$ within the distance *B* in $P_{rock}$). As shown in Equation (Equation 28), the local normal saliency of point $c_{i,j}\left[k\right]$ is
(28)$$D\left(c_{i,j}\left[k\right]\right)=\left(c_{i,j}\left[k\right]-S_{i}\left[k\right]\right)\cdot N\left(S_{i}\left[k\right]\right)$$In order to ensure the integrity of steel arch extraction and reduce the impact of noise, the candidate points were sorted by their local normal saliency, and the most salient points were chosen to be added into the point cloud $P_{o}\left(i\right)\left[k\right]$. The maximum number of elements in a salient point cloud $P_{o}\left(i\right)\left[k\right]$ is restricted to 2B/voxelsize.Since there may be no other points around the seed point, there are two scenarios. In Equation (Equation 29), the interpolation of the missing point cloud is realized by directly assigning the seed point to the optimum point.
(29)$$\left\{\begin{array}{ccc} O_{i+1}\left[k\right] & =mean(P_{o}\left(i\right)\left[k\right]), & P_{o}\left(i\right)\left[k\right]\neq \emptyset \\ O_{i+1}\left[k\right] & =S_{i}\left[k\right], & P_{o}\left(i\right)\left[k\right]=\emptyset \end{array}\right.$$The detailed interpolating effect of optimum points *O* are shown in Figure 15.

Essentially, the process of seed point growth is a fast convergence to the optimal local solution. Figure 16 shows that the evenly distributed initial seed points converge fast to the nearest steel arch and grow along the arch until they reach the other side of the tunnel.

As shown in Figure 17, the proposed method is explicit and effective in detecting steel arches in both square and round tunnels. The square tunnel point cloud used to test the DEG method is obtained by deforming the circular tunnel data. Therefore, it can be proved that the DEG method is not sensitive to the geometrical outline of the tunnel.

## 4. Experiment

In order to evaluate the proposed arch detection method, a series of experiments were conducted under real scenarios. Since there was no tunnel-specific method for arch detection that could be used for the control experiment, some feasible methods introduced in Section 3.4.1 were selected, including the profile radius method [37], NARF [40], boundary detection [10], and region-growing. Since the arch point cloud is a feather-edged layer of the point cloud, a Robust Feature-Preserving Denoising (RFPD) method [48] was used to denoise the wire mesh points in $P_{rock}$ and preserve the sharp and fine-scale 3D features of arches for NARF [40].

Traditional methods require testing and adjustment of parameters by experimenters. For these methods, the ideal feature point recognition effect of each method was obtained by adjusting parameters manually. The parameter settings of the proposed algorithm have been summarized in Table 5. They were set based on actual conditions, without manual parameter regulation.

The Industrial PC we used was configured with CPU i7-6820EQ and DDR4 32G. The experimental results of these methods on the same group of real tunnel point clouds ID 1 (Section 2, Table 3) are shown in Figure 18. The comparison of parameter settings and running results of these six methods are shown in Table 6.

The proposed algorithm can realize an adaptive threshold in this problem scenario. The parameter settings used in each step of the proposed algorithm that needed to be set according to construction standards in advance were based on the three parameters of voxelsize, $W_{a}$ and *B*, which are given in Section 2. Therefore, the parameter settings can meet the actual requirements of engineering applications.

### 4.1. Qualitative Analysis

There were some difficulties in steel arch extraction, including: (a) Some steel arches installed askew; (b) Holes and defects of the point cloud caused by occlusions; (c) Interference caused by the rock surface with complex shapes; (d) Extracting the steel arches having been covered by concrete; (e) Interference caused by the similarity of geometric characteristics between wire mesh and steel arches.

Combined with the Figure 18 and Table 3, we analysed the experimental results in detail in Table 7. The symbol ‘*√*’ in Table 7 means that the method can avoid the interference, while ‘×’ means the opposite. As can be seen from Figure 18 and Table 7, our method has more obvious advantages over other methods.

### 4.2. Quantitative Analysis

To quantify the performance of the steel arch extraction algorithm, the boundary of the steel arch point clouds of the pre-processed data introduced in Section 2 was labelled manually and compared with the automatic detection results. The incomplete area of steel arch and the interpolation area of steel arch point cloud in the point cloud were not considered. The variables TP (true-positive) and FP (false-positive) respectively mean the number of points were labelled as steel arch points correctly and incorrectly, and FN (false-negative) means the number of points were falsely labelled as non-steel arch points. The precision, recall, and F−Score criteria used by Yang et al. [49] was adopted to evaluate the steel arch extraction results:(30)$$Precision=\frac{TP}{TP+FP};Recall=\frac{TP}{TP+FN};F-Score=2\cdot \frac{Precision\cdot Recall}{Precision+Recall}$$

This experimental result is the cumulative result obtained after tunnel orientation calibration, rock surface segmentation, and steel arch detection of the pre-processed point cloud. The manually labelled data were based on the pre-processed data and only used to evaluate the extraction effect of the algorithm on the edge of the steel arches. Therefore, the voxel filtering method adopted in the pre-processing had no direct effect on the precision and recall rate of steel arch identification.

From an upward view inside the tunnel, Figure 19 shows the arch-extraction result of the proposed algorithm compared with the manual annotation data.

Table 8 shows the performance of the proposed algorithm in terms of the above-mentioned two criteria for steel arch extraction.

The experimental result shows that the average precision of the algorithm reaches 92.2%, and the automatically labelled arch points are evenly distributed alongside the arch edges. Since DEG has a specific step size, some steel arch points were skipped during the growing process. By recalling the salient point cloud $P_{o}$, the average recall and F−score rate of the proposed algorithm reached 89.1% and 90.6%, respectively.

Furthermore, we compared the average precision, recall, and F−Score rates of the results obtained by different methods, and obtained intuitive comparison results, as shown in Table 9. According to the experimental results, the method proposed in this paper had a higher precision and recall rate than that of the control group.

### 4.3. Anti-Noise Performance

In order to estimate the anti-noise performance of the proposed algorithm, we added Gaussian noise (expected value μ = 0) to the pre-processed point cloud ID 1 (Section 2, Table 3) and processed the noise mixed data. The experimental result of a series of standard deviations σ (σ = 10, 50, 100, 150, 200, 300, 500, 800, 1200 mm) is shown in Figure 20. Since the thickness of the steel arch was 200 mm, the profile of the steel arch edge was completely fuzzy on the condition that σ≥ 200 mm. However, the results of σ = 200, 300 mm in Figure 20 show that the proposed algorithm was still able to identify the edge of the steel arches.

The results of σ = 500, 800 mm in Figure 20 show that the proposed algorithm was still able to identify some regions of steel arches. When $\sigma >W_{a}$, the point clouds between multiple steel arches were completely blurred, and the results lost the reference value.

In conclusion, the proposed algorithm appears to be robust to the noise, especially when σ≤B.

## 5. Conclusions and Future Work

For automatically extracting the steel arch installed on the complex rock surface, this paper presents a novel algorithm to extract a tunnel steel arch. In this paper, steel arches were extracted from nine groups of LiDAR point clouds collected in the tunnel. The point clouds mainly consisted of rock surfaces, working surfaces, and shotcrete surfaces. The specific parameters are shown in Section 2. Firstly, a refine function for correcting the tunnel point cloud axis by minimizing the Rotational Projection Density Variance (RPDV) was proposed. Secondly, the rock surface was segmented from the tunnel surface in the region-growing and DBSCAN method with the parameters obtained by the Tunnel Section Sequence Analysis (TSSA). Finally, a Directed Edge Growing (DEG) method based on the region-growing principle was proposed to detect multiple steel arches on the rock surface in the tunnel. The experimental results show that the proposed algorithm can effectively detect multiple steel arches of the tunnel without manual assistance. The precision rate, recall rate, and F-score of the quantitative experiment reached 92.1%, 89.1%, and 90.6%, respectively. Besides, the proposed algorithm seemed to be robust to the tunnel profile and the incomplete data. The detection algorithm of steel arch has important engineering application significance, since it can be used to provide constraint information for planning the shotcrete path and detecting the precision of the steel arch installation automatically.

There is some research on tunnel steel arches extraction that can be carried out in the next stage. We will further study the steel arches extraction algorithm for more complicated tunnels, such as tunnels excavated using the multi-step excavation method [50]. A comprehensive assessment system for the precision of arch steel extraction needs to be established in the future. Besides, the registration problem of mobile laser scanning data to identify steel arches can also be studied further.

## Figures and Tables

**Figure 1 sensors-19-03972-f001:**
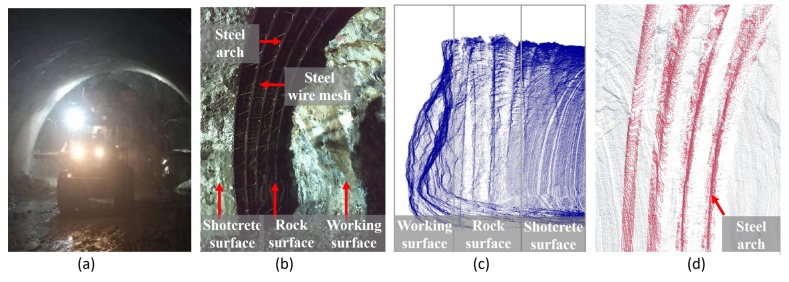
The support structure in the tunnel. (**a**) The environment of shotcrete in the tunnel. (**b**) The structures of the initial shotcrete area. (**c**) Point cloud of steel arches. (**d**) Point cloud of three kinds of surfaces.

**Figure 2 sensors-19-03972-f002:**
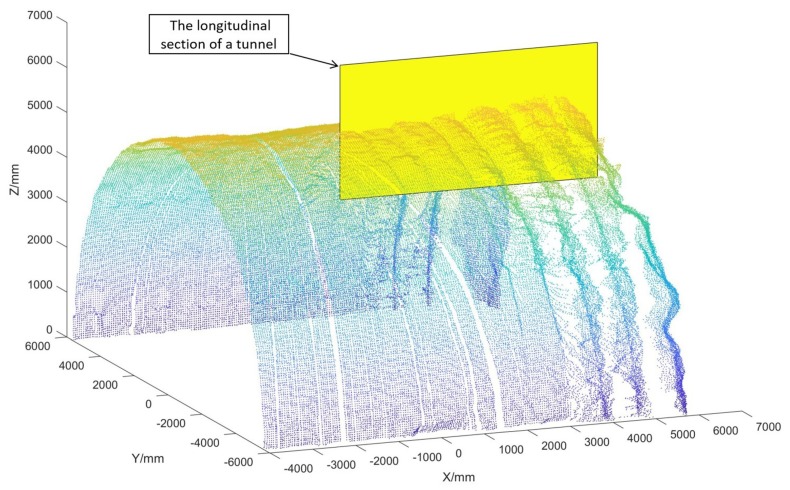
A longitudinal section taken from the tunnel point cloud.

**Figure 3 sensors-19-03972-f003:**
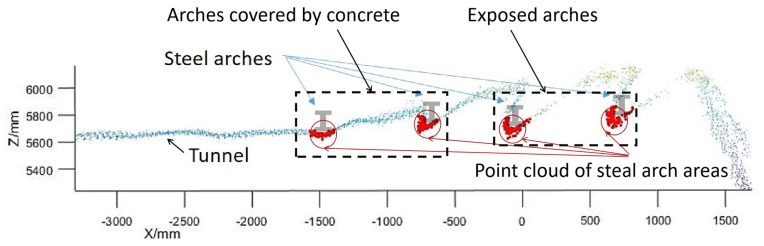
Steel arch areas on the longitudinal section of a tunnel. The red point clouds are the section of the arch areas.

**Figure 4 sensors-19-03972-f004:**
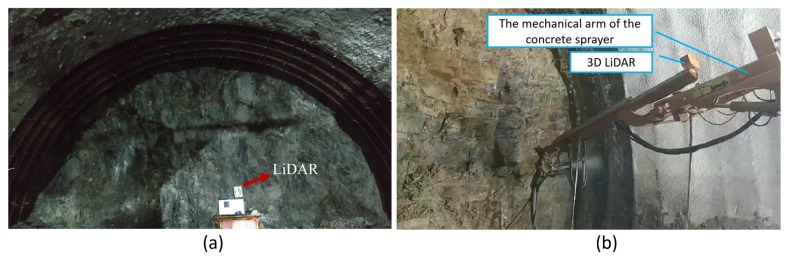
The LiDAR in the real tunnel. (**a**) Data collection site; (**b**) practical application.

**Figure 5 sensors-19-03972-f005:**
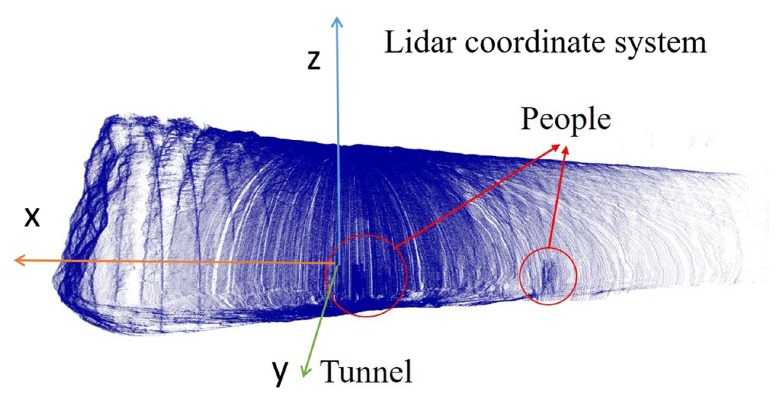
Original point cloud.

**Figure 6 sensors-19-03972-f006:**
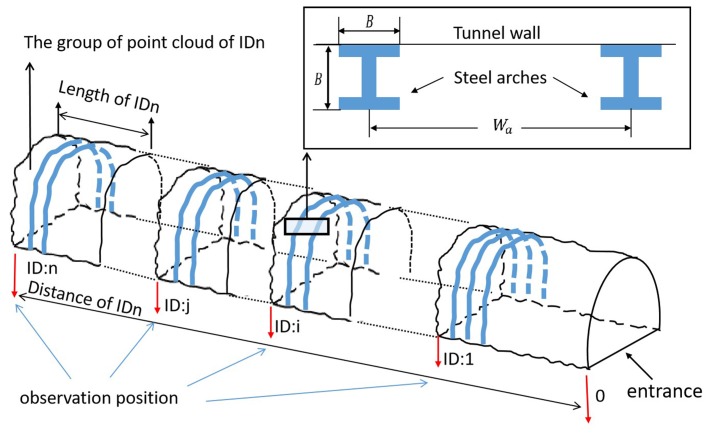
Parameters of the tunnel data set.

**Figure 7 sensors-19-03972-f007:**
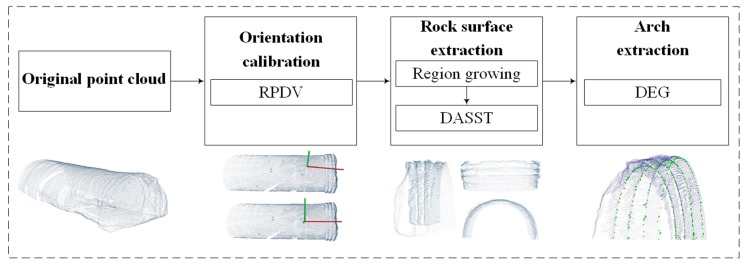
The overview of the tunnel steel arches extraction.

**Figure 8 sensors-19-03972-f008:**
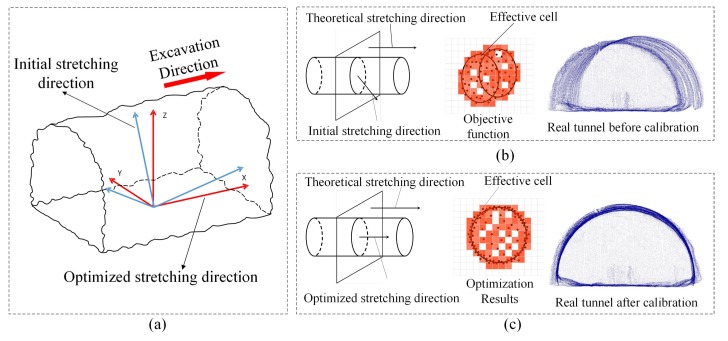
The theory of orientation calibration. (**a**) Initial and optimized stretching direction of the tunnel. (**b**) Schematic diagram of the tunnel before the tunnel axis calibration. (**c**) Schematic diagram of the tunnel after the tunnel axis calibration.

**Figure 9 sensors-19-03972-f009:**
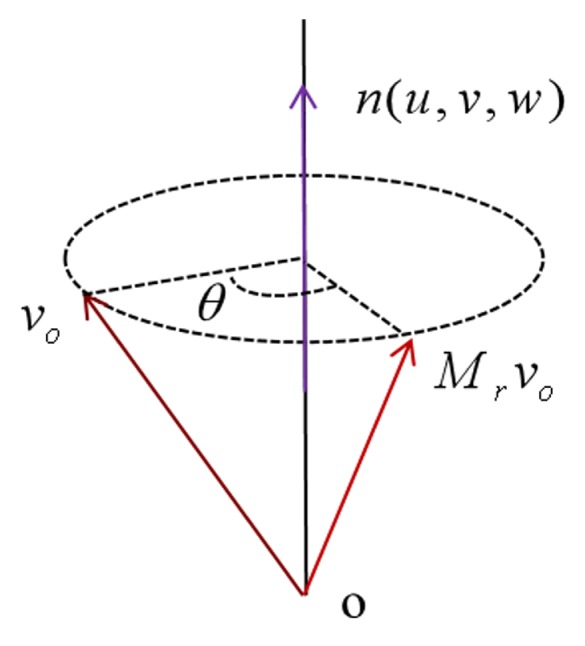
Geometrical interpretation of the rotation formula $M_{r}\left(n,\theta \right)$.

**Figure 10 sensors-19-03972-f010:**
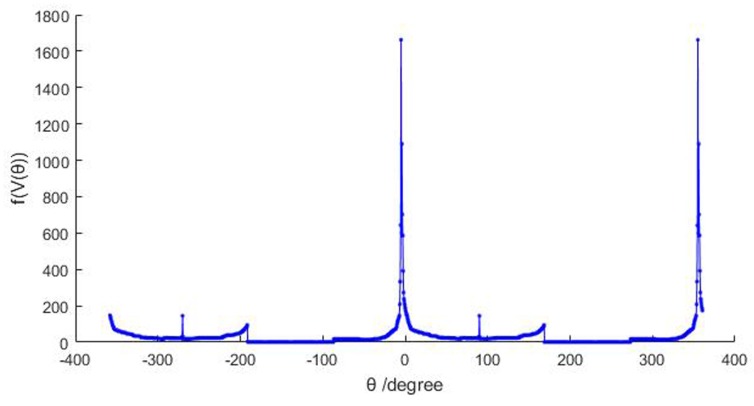
Projection density f(V(θ)) varies with rotation angle θ (using the data ID1 in Table 3).

**Figure 11 sensors-19-03972-f011:**
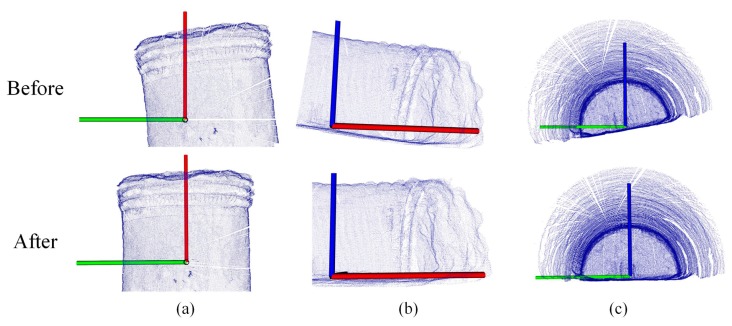
Comparison of the tunnel point cloud before and after calibration: (**a**) Top view; (**b**) side view; (**c**) front view.

**Figure 12 sensors-19-03972-f012:**
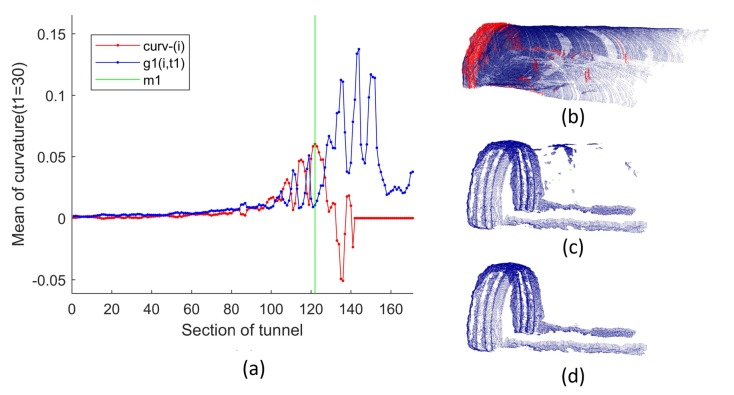
The process of segmenting the rock surface. (**a**) Differential Analysis for the Section Sequences of the Tunnel point cloud (DASST) result of the tunnel (using the data ID1 in Table 3 and $W_{d}=100mm$). DASST used for $P_{tuns}$: Sequence $\bar{curv}$(i) is the mean curvature of the tunnel sections; $g_{1}\left(i,t_{1}\right)$ is the discrete sliding cumulative function with step size $t_{1}$; $m_{1}$ is the optimal solution of $g_{1}\left(i,t_{1}\right)$. (**b**) region-growing; (**c**) removing the largest surface; (**d**) removing the discrete points.

**Figure 13 sensors-19-03972-f013:**
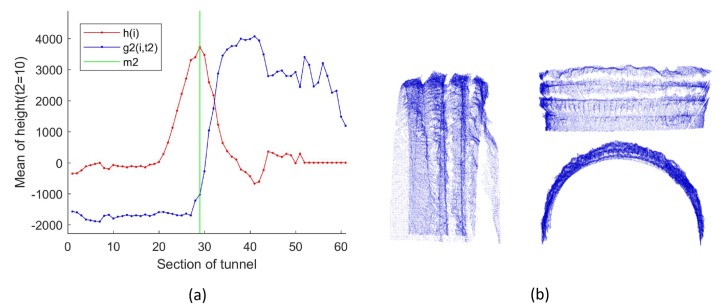
DASST result of the tunnel (using the data ID1 in Table 3 and $W_{d}=100mm$). (**a**) DASST used for $P_{tuns^{\prime }}$: Sequence h(i) is the mean height of the tunnel sections; $g_{2}\left(i,t_{2}\right)$ is the discrete sliding cumulative function with step size $t_{2}$; $m_{2}$ is the optimal solution of $g_{2}\left(i,t_{2}\right)$; (**b**) three views of segmentation results of the rock surface.

**Figure 14 sensors-19-03972-f014:**
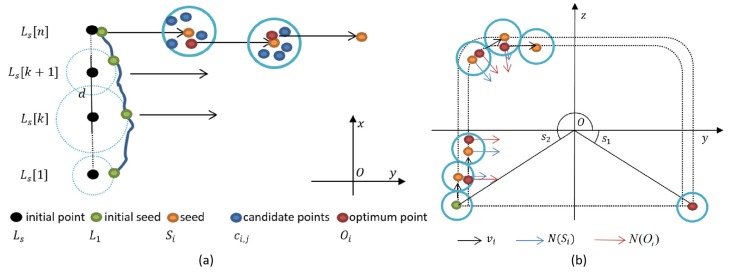
Directed Edge Growing (DEG). (**a**) Edge growing process in the top view; (**b**) edge growing process in the main view .

**Figure 15 sensors-19-03972-f015:**
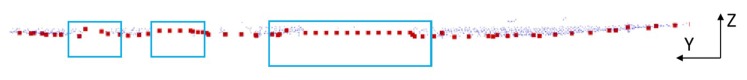
Interpolating the missing points.

**Figure 16 sensors-19-03972-f016:**
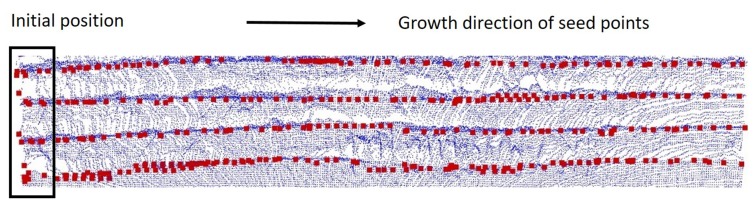
Completed recognition result of the optimum points, *O*.

**Figure 17 sensors-19-03972-f017:**
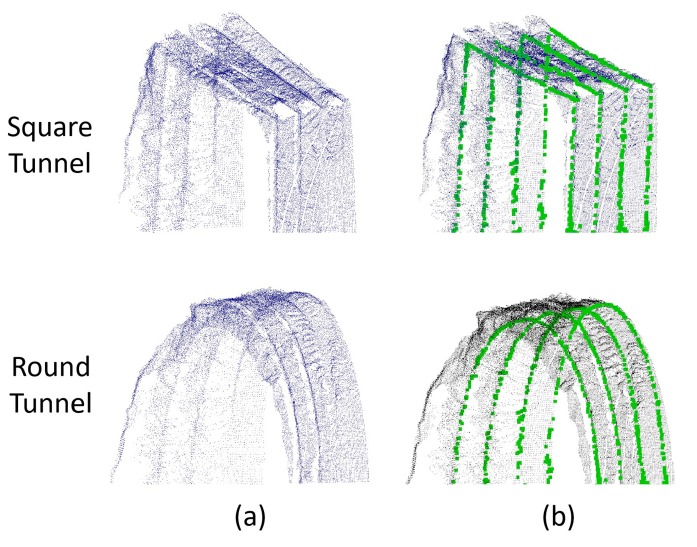
Arch extraction results of both round and square tunnels. (**a**) The point cloud of the rock surface ($P_{rock}$); (**b**) the arch extraction results of $P_{rock}$.

**Figure 18 sensors-19-03972-f018:**
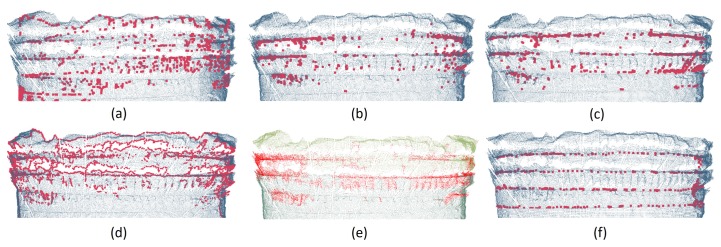
Qualitative experimental results comparison. (**a**) Profile radius; (**b**) NARF; (**c**) RFPD+NARF; (**d**) Boundary detection; (**e**) region-growing; (**f**) the proposed method.

**Figure 19 sensors-19-03972-f019:**
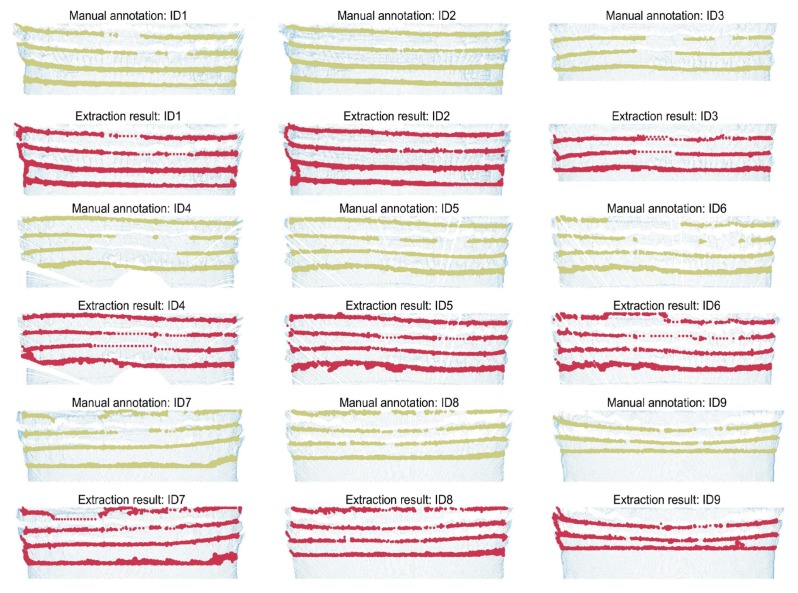
Arch extraction result of the proposed algorithm compared with the manual annotation data from an upward view.

**Figure 20 sensors-19-03972-f020:**
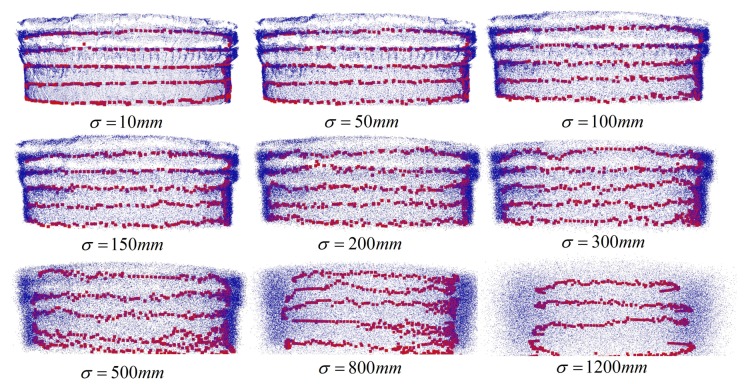
The extraction effect of the proposed arch extraction of the tunnel point data added with Gaussian noise.

**Table 1 sensors-19-03972-t001:** Comparison of data acquisition methods.

Technical Specifications	Advantages	Limitations
Total Station	High accuracy; Easy to locate	Time-consuming; Sparse points
RGB cameras	Dense and ordered point cloud	Insufficient light source in tunnels
LiDAR	High accuracy; Dense point cloud; Less impact from harsh environment	Non uniform point cloud

**Table 2 sensors-19-03972-t002:** Description of the 3D scanning device.

Laser Scanner	Scanning Angular Increment	Scan Range	Scanning Frequence	Rotary Table	Rotation Angular Increment	Absolute Accuracy of the LiDAR
P+F R2000 UHD	0.05${}^{\circ }$	30 m	50 Hz	PT-GD201	0.05${}^{\circ }$	±25 mm

**Table 3 sensors-19-03972-t003:** Tunnel point cloud data.

ID	Number of Points	Number of Pre-Processed Points ${}^{a}$	Length (mm) ${}^{b}$	Observation Position (m) ${}^{c}$	$W_{a}$ (mm)	*B* (mm)
1	3,741,570	538,935	28,247	75.0	1000	200
2	3,858,638	555,716	27,305	77.5	1000	200
3	1,616,947	232,991	17,720	82.4	1000	200
4	1,439,603	207,476	18,289	87.8	1000	200
5	1,546,882	222,809	17,786	95.6	1000	200
6	1,600,717	230,533	19,756	100.5	1000	200
7	2,240,123	322,657	16,660	105.9	1000	200
8	1,709,192	246,209	20,360	109.8	1000	200
9	2,265,829	326,387	17,913	121.3	1000	200

${}^{a}$ The voxel size is 26 mm. ${}^{b}$ The length of the group point cloud along the *X*-axis. ${}^{c}$ The observation position relative to the exit of the tunnel.

**Table 4 sensors-19-03972-t004:** Comparison of the theory and limitations of different methods.

Method	Method Principle	Limitations
Tunnel axis + Profile Radius [37]	Comparing the difference between the distance from the real arch profile to the tunnel axis and the distance from the standard arch profile to the tunnel axis.	The method is sensitive to the interference resulted from the steel mesh as well as the errors in the tunnel axis calibration, and the arch installation are inevitable.
Harris3D [38], SIFT3D [39]	These feature points were extended from the feature description method of 2D images, and are widely used for point cloud registration, recognition, and classification.	They are not applicable to distinguish steel arches from steel grids since steel arches arranged longitudinally and steel grids arranged horizontally have similar Harris3D and SIFT3D characteristics.
NARF [40]	The method can be used to take the center of the tunnel point cloud as the observation point and expand it into a range image for edge detection.	The recognition effect of the NARF method is unstable and needs to be improved.
Boundary detection [10]	Based on the given Euclidean distance and k-tree search method, the boundary of the hole is detected after the point cloud is triangulated.	The shielding effect of steel arches on laser results in multiple types of banded holes in the point cloud behind the arch.
region-growing	The seed points keep growing according to the characteristics of the surface until the seed points reach the boundary.	The segmentation effect depends on the given parameters and has poor adaptability to rough and complex surfaces.
MVCNN [41], GVCNN [42]	The 3D point cloud is projected into 2D images from multiple views, and CNN is used to extract features for each view in combination with the image processing method.	The projection method will lead to the loss of the key geometric spatial information of the arch structure, which will affect the segmentation accuracy of the point cloud.
Voxnet [41], PointGrid [44]	The disordered point cloud is voxelized into a regular structure, and then the neural network architecture is used to learn its characteristics.	Low efficiency of voxel grid arrangement; large memory occupied in the calculation process; time consuming; information loss.
Pointnet [41]	This method extracts the feature description of each independent point and the description of global point cloud features. Therefore, the point cloud of the steel arch area should be segmented into independent individuals to form a data set.	The relationship between points and neighborhood information is not considered, resulting in information loss when dealing with large-scale point cloud data. It can be used to detect the areas instead of the edge of the steel arches.

**Table 5 sensors-19-03972-t005:** Summary of the parameter-setting of the proposed algorithm.

Step	Parameter	Meaning
RPDV	*B*	The grid size of RPDV
DASST	$W_{a}$	Radius used for calculating curves and normal vectors
DASST	$W_{d}\in \left[voxelsize,W_{a}\right]$	The slicing thickness of DASST
DASST	$3W_{a}/W_{d}$	The step size used for region-growing threshold
DASST	$W_{a}/W_{d}$	The step size used for segmentation after DBSCAN
DEG	*B*	Searching radius used for candidate points
DEG	*B*	Interval distance of initial points
DEG	2B	The step size used for DEG
DEG	2B/voxelsize	Maximum number of elements in point cloud $P_{o}\left(i\right)\left[k\right]$

**Table 6 sensors-19-03972-t006:** Method comparison.

ID	Method	Times	Parameter
1	Profile Radius [37]	5.517 ms	Radius = Rs + 630→640
2	NARF [40]	6.179 ms	Search Radius = 100 mm
3	RFPD [48] + NARF [40]	19.667 ms	Search Radius = 100 mm
4	Boundary detection [10]	4.392 ms	Search Radius = 100 mm Normal Radius = 100 mm
5	region-growing	7.198 ms	Curve threshold = 0.03 Normal threshold = 30
6	The proposed method	12.178 ms	$W_{a}$ = 1000 mm, *B* = 200 mm,
	(RPDV + DASST + DEG)		voxelsize = 26 mm

**Table 7 sensors-19-03972-t007:** Analysis of resistance to interference factors.

Difficulty	Profile Radius	NARF	RFPD + NARF	Boundary Detection	Region-Growing	Ours
The steel arch is askew	×	*√*	*√*	*√*	*√*	*√*
Point cloud holes and defects	*√*	*√*	*√*	×	*√*	*√*
Rocks of complex shapes	*√*	×	×	*√*	×	*√*
Steel arches covered with concrete	*√*	×	×	×	×	*√*
The similar geometric characteristics of wire mesh and steel arch	×	×	×	×	×	*√*

**Table 8 sensors-19-03972-t008:** Assessment of the steel arch point clouds extraction results.

ID	1	2	3	4	5	6	7	8	9	Average
Precision	0.907	0.926	0.919	0.913	0.928	0.924	0.923	0.941	0.909	0.921
Recall	0.914	0.901	0.918	0.907	0.875	0.899	0.855	0.904	0.843	0.891
F-Score	0.910	0.913	0.919	0.910	0.901	0.911	0.888	0.922	0.875	0.906

**Table 9 sensors-19-03972-t009:** Comparison of precision and recall rate of each method.

Method	Precision	Recall	F-Score
Profile Radius [37]	0.125	0.053	0.074
NARF [40]	0.218	0.424	0.288
RFPD [48] + NARF [40]	0.289	0.613	0.393
Boundary detection [10]	0.081	0.572	0.142
region-growing	0.064	0.559	0.115
The proposed method	0.921	0.891	0.906

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
