# Peer review of "Automatic Tunnel Steel Arches Extraction Algorithm Based on 3D LiDAR Point Cloud"

_sensors, 2019, doi:10.3390/s19183972_

Round 1

Reviewer 1 Report

Indeed, the improvement of revised version is significant and many of the critical points have been well addressed. Based on the all the progress made in this version, I will advise and suggest, given by “From reviewer”, some further points to be refined or corrected to justify the value of this work following the authors’ responses.

(The low recall ratio tells the weakness in the completeness of steel arches extraction. )

Response:

Thanks for the reviewer's suggestion, we changed the original algorithm to improve the detection recall rate.

The original method only selects one seed point in each candidate area, which is not reasonable enough. Single seed spot is susceptible to noise and reduces the recall rate. We added an optimal seed point cloud Po in P15 I314. Po reduced the interference of noise on the selection of seed points and significantly increased the recall rate.

The rough extraction method for steel arch extraction was dismissed after the authors' consideration. The seed points needed in the rough extraction process were obtained by voxelization of the point cloud. The rough extraction can compress the point cloud, retain the steel arch profile features, prevent DEG process from deviation, and shorten the time of DEG. However, this method will neglect some steel arch points. By dismissing the rough extraction process, the recall rate was improved, and the possibility of cumulative errors was reduced.

From reviewer: Since much improvement in recall rate has been achieved as compared to the previous version and also to the related works, it would be meaningful and useful to show how the extracted steel arches render the true steel arches. To this end, points of the extracted and true steel arches can be coded with different colors for a clear comparison. Besides, color images clearly showing the appearance of steel arches are helpful for visual impression on extraction performance.

2. Response to comment:

The geometric accuracy of the point clouds has not been evaluated along with the extraction processing considering the usage of low-cost laser scan instrument.

Response:

Considering the reviewer's suggestion, we added a geometric accuracy analysis of the low-cost laser scan instrument in Section 2. The geometric error was produced in the process of the voxelization in pre-processing. According to the voxel size, the geometric accuracy is estimated to be about 25mm.

From reviewer: The interval of points is not equivalent to the geometrical accuracy of the LiDAR point. The geometrical accuracy of LiDAR point has to examine the coordinate accuracies of points, which are affected, among others, mainly by ranging and angle errors and also varies among employed instruments.

Author Response

Response to Reviewer 1 Comments

Point 1: Since much improvement in recall rate has been achieved as compared to the previous version and also to the related works, it would be meaningful and useful to show how the extracted steel arches render the true steel arches. To this end, points of the extracted and true steel arches can be coded with different colors for a clear comparison. Besides, color images clearly showing the appearance of steel arches are helpful for visual impression on extraction performance.

Response 1: Thanks for the reviewer's suggestion. In the new version of the manuscripts, we added Figure17 to clearly show the appearance of the steel arch. In Figure17, points of the extracted and true steel arches was coded with different colors for a clear comparison.

Point 2: The interval of points is not equivalent to the geometrical accuracy of the LiDAR point. The geometrical accuracy of LiDAR point has to examine the coordinate accuracies of points, which are affected, among others, mainly by ranging and angle errors and also varies among employed instruments.

Response 2: Thanks for the reviewer's reminding. The ranging and angle errors are inherent properties of measuring instruments. This paper mainly studies the detection effect of the algorithm for feature detection of the existing cloud data. Therefore, the measurement error of the measuring equipment was not further analyzed and the measurement precision of the equipment was directly taken as the known equipment parameter. We have modified the relevant statements in Section2.3 (Acquisition equipment) and Section2.4 (Pre-processing of the data), and added information of the equipment accuracy in Table 3.

Reviewer 2 Report

First of all, excuse-me for any English mistakes since I’m not an English native speaker.

In my opinion the paper has potential to be published, but not in the present form. There are some points that must be revised by the authors. Concerning the writing style, in some parts the authors mention some details of the method before the detailed method explanation and mathematical development and, in my opinion, this style harmed to clearly understand some parts of the paper. Follow attached a PDF file with some comments to the authors and editors indicating some points to be revised.

In the attached PDF file some references are suggested and in the sequence there are some other papers related to the treated subject:

DU, L.; ZHONG, R.; SUN, H.; LIU, Y.; WU, Q.  Cross-section Positioning Based on a Dynamic MLS Tunnel Monitoring System. The Photogrammetric Record, 9 ago. 2019. DOI: 10.1111/phor.12287

LAM, S. Y. W.  Application of terrestrial laser scanning methodology in geometric tolerances analysis of tunnel structures. Tunnelling Underground Space Technology, 21 (3), 410, 2006.

ATTARD, L.; DEBONO, C. J.; VALENTINO, G.; DI CASTRO, M.  Tunnel inspection using photogrammetric techniques and image processing: A review. ISPRS Journal of Photogrammetry and Remote Sensing, v. 144, p. 180–188, 2018. DOI: 10.1016/j.isprsjprs.2018.07.010

Round 2

Reviewer 1 Report

Response 1: Thanks for the reviewer's suggestion. In the new version of the manuscripts, we added Figure17 to clearly show the appearance of the steel arch. In Figure17, points of the extracted and true steel arches was coded with different colors for a clear comparison.

It should be Figure 19, instead of Figure 17.

All raised question points have been satisfactorily responded.

Author Response

Point 1: It should be Figure 19, instead of Figure 17. All raised question points have been satisfactorily responded.

Response 1: Thanks for the reviewer's reminding and comments. It is Figure19 that shows the appearance of the steel arch coded with different colors for a clear comparison.

Reviewer 2 Report

Again, excuse-me for any English mistakes since I’m not an English native speaker.

The authors reviewed the paper, improving many indicated points. Follow attached one pdf file in which I indicate some points to be verified and reviewed by the authors.

After these modifications, I consider that the paper can be accepted.

Author Response

This manuscript is a resubmission of an earlier submission. The following is a list of the peer review reports and author responses from that submission.

Round 1

Reviewer 1 Report

/* Layout-provided Styles */ div.standard { margin-bottom: 2ex; }

Paper title: Automatic Tunnel Steel Arches Extraction Algorithm Based on 3D LiDAR Point Cloud

Authors : Wenting Zhang, Wenjie Qiu, Di Song and Bin Xie

Paper overview: This paper proposes an automatic steel arch detection and extraction framework for processing tunnel LiDAR scans. The framework is structured in three subsequent procedures: i) tunnel axis calibration, ii) rock surface segmentation and iii) arches extraction. Each procedure is formalized and detailed. The framework is evaluated in full scale conditions in terms of precision/recall scores. The results are compared to five different methods.

Review:

The proposed methods and the conducted investigation are interesting. The addressed application provides a particular environment for LiDAR based mapping.

Hereafter, some comments and questions related to this work:

1. Fig. 2 shows steel arch areas of a tunnel section. This figure has no convention of the illustrated elements. What do represent red points in the figure?

2. In Fig. 5, a point cloud example of a tunnel section is illustrated. It is not clear how multiples scans were aligned and how the LiDAR observation location and attitude was measured/estimated. This point must be clearly addressed since it is of key importance to quantify the impact and the robustness of alignment errors on the arch extraction procedure.

3. In Sec 3.3, p5 l134, it is mentioned that a voxel filtering is applied to the point cloud as part of a data pre-processing. It is not mentioned how an appropriated voxel size was selected. Please provide a error impact quantification of a wrong voxel size choice regarding precision/recall indicators of the arc extraction.

4. In p5, l137, it is stated that DBSCAN is intended to retrieve the point cloud of the main par of the tunnel. What does precisely mean the main part of the tunnel?

5. How many classes are set for Kd-tree on DBSCAN? How is the seed point set ? Please explain.

6. RPVD is the final step of the alignment of the point cloud. This contributed method consists in optimizing the point cloud orientation which minimizes the dispersion of the point cloud 2D projection. In p7, l175 it seems that \theta is optimized by introducing angle variations. How is warranted the optimality of \theta? How local minima is avoided during this optimization?

7.Fig. 10 illustrates mean curvature and mean height estimates employed for determining the optimal mean curvature and height of the tunnel, m1 and m2 respectively. How m1 and m2 values ware estimated? This point merits to be clarified.

8. Arch segmentation was evaluated and compared using precision/recall scores. Results do not represent any improvement concerning precision score. Recall score of the proposed method certainly is improved with respect to other approaches. Regarding the number of classified points (298 in Table 4) and considering the equation 38, the approach provides a high number of false negatives (i.e. missed detections). I think that the proposed approach remains sensible to noise and to accumulated errors of data pre-processings. This issue must be deeply analyzed.

Typos :

Several typos and grammar errors were identified in the paper. Some of them are (not exhaustive list):

* P3, l60 Hanet al.

* P7, l157 The the

* P7, l171 v is a vector

* P7, l157 Rodrigues' rotation

* P8, l189 largest amount of points

* P9, before Eq. 19, 3W/100 -> 3Wa/100

* P11, l217 used to took

* P11, l219 additioon

* P11, l220 detection[36]can

* Fig 13 (a) -> (b)

* P16, l120 parameters Settings

* P17, l356 missing reference

* P17, l363 the precise of the algorithm (precise is not a noun)

Reviewer 2 Report

The authors designed an effective workflow for automatic extraction of tunnel steel arches from 3D LiDAR point cloud. The results show the high precision of detected targets through this proposed approach.

Yet, the low recall ratio tells the weakness in the completeness of steel arches extraction. Besides, the geometric accuracy of the point clouds has not been evaluated along the extraction processing considering the usage of low-cost laser scan instrument. Several other technical issues and writing quality are advised below for authors’ reference:

1. It is not quite clear about how to determine the stretching direction in section 3.3.

2. Inconsistent writing tense makes the manuscript a low readability.

3. Some equations are lack of explanation in formulae or in notations.

4. Examples of typing error are marked and some questions are given in the attached PDF.

Reviewer 3 Report

The article presents an interesting premise regarding 3D point clouds recognition and extraction for automation. The followings are comments to further improve the current state of the manuscript.

1.      There have been many prior scholarly works on ‘point cloud object detection with multi-view convolutional network’. Please, create tables or figures for literature reviews, as opposed to listing all references one by one. Then, the authors should (a) highlight their limitations; (b) clarify why the authors’ work is significant; and (c) elaborate what gap-in-knowledge in the underlying theories and current practices can be filled through this work.

2.      In the case of tunnels, it is not easy to acquire data, especially in the registration process. As described by the authors, it is necessary to study the matching problem in order to improve the field applicability. It would be better to briefly describe the process and limitations of data acquisition. (ex: 1.1 Data Acquisition methods, or 3. Methodology)